# Undenatured Type II Collagen (UC-II) in Joint Health and Disease: A Review on the Current Knowledge of Companion Animals

**DOI:** 10.3390/ani10040697

**Published:** 2020-04-17

**Authors:** Hasan Gencoglu, Cemal Orhan, Emre Sahin, Kazim Sahin

**Affiliations:** 1Department of Biology, Faculty of Science, Firat University, Elazig (+90) 424, Turkey; hgencoglu@firat.edu.tr; 2Department of Animal Nutrition, Faculty of Veterinary Medicine, Firat University, Elazig (+90) 424, Turkey; corhan@firat.edu.tr (C.O.); esahin@bingol.edu.tr (E.S.)

**Keywords:** collagen, inflammation, joint degeneration, osteoarthritis, pain

## Abstract

**Simple Summary:**

Osteoarthritis (OA), the most common joint disease affecting humans and animals, is a painful, degenerative, and inflammatory disease that affects synovial joints and ultimately leads to loss of mobility. Non-pharmacological preventive approaches, several pharmaceutical therapeutic agents, and some medicines may reduce the progression of OA in animals. Many clinical and experimental studies have revealed that the undenatured form of type II collagen (UC-II) offers common health benefits to patients with OA.

**Abstract:**

OA is quite common in companion animals, especially in large breed dogs and horses. Collagen, the most abundant protein of mammals, has specific connective tissue types for skin, bones, reticulate, basal lamina, bones, cell surfaces, while type II collagen (UC-II) forms the main structure of cartilage tissue. Even at the smaller dosages, UC-II has also been reported to be more effective than the glucosamine and chondroitin sulfate supplements, which are the supplements most frequently used in the market. In this review, we summarize the effects of UC-II on joint health and function in health and disease conditions in companion animals.

## 1. Introduction

Osteoarthritis (OA), also known as a degenerative joint illness, is a chronic, painful, and inflammatory disease that affects the joints in knees, feet, hips, spine, subchondral bone, synovial membranes, and periarticular tissues, and leads to immobility and morbidity in humans, dogs, horses, and cats, and companion animals throughout the world [1,2,3]. It is characterized by chronic joint pain, stiffness, inflexibility, swelling, narrowing of joint spaces, and formation of osteophytes and lameness [1,2,3,4,5,6]. Mobility reduction and pain caused by OA have a negative effect on the quality of life, comfort level, walking, exercise tolerance, activity, urinary and fecal habits behavior in animals [6,7,8]. It is known that nearly 20% of canine pets spontaneously develop osteoarthritis, and this translates to at least 15 million dogs in the USA alone [6]. OA mostly affects the large breed dogs [9], geriatric cats [10], and sport horses [11]. OA occurs in the dog populations mostly because of overweight and/or obesity, insufficient exercise, injury, becoming old, having an infection, immune disorders, or genetic predisposition [12]. Dogs also suffer more often with OA than immune-mediated arthritis [13]. Epidemiological studies have reported that risk factors for the development of OA comprise aging, excessive or non-physiological burdens, obesity, traumas, hormonal ailments, or a mixture of several factors [14,15,16]. Even though the exact etiology of the OA has not been identified yet, in the long run, articular cartilage degeneration results in changes to both as metabolisms of chondrocytes and synoviocytes such that inflammatory cytokines that are formed damage the ability of chondrocytes to renovate the cartilage matrix [8,13,17].

The main goal in OA management in animals is to control clinical findings by protecting the joints from OA, reducing pain, increasing mobility, and, therefore, the quality of life [18]. For this purpose, non-pharmaceutical treatment options include surgery, weight loss, exercise modification, and physical therapy [19]. Surgical intervention or arthroplasty is frequently used to prevent joint changes or restore its function, but there have been no gold standards for both targets yet [8]. There are also pharmaceutical treatment options. For example, glucosamine, chondroitin, undenatured form of type II collagen, pentosane polysulphate, avocado/soybean unsaponifiables, green-lipped mussel, milk protein, creatine, and amino acid represent the largest category of natural supplements for veterinary medicine [16].

The undenatured form of type II collagen (UC-II), a nutritional supplement, is derived from chicken sternum cartilage and is a powdered, glycosylated, and shelf-resistant component [17]. Previous studies have revealed that UC-II reduced lameness after general pain, pain during limb manipulation, and physical exertion in arthritic dogs given 4 mg or 40 mg of UC-II daily [20]. Currently, only limited reviews on the effects of UC-II in animal OA have been published [21]. The aims of this review are to summarize the scientific data available in the literature on UC-II evaluated in animals, including dog, horse, and cat OA, and to discuss some studies about how to improve several aspects of research and issues with UC-II supplements, such as bioavailability and molecular mechanisms. In addition, companion animal studies related to UC-II are presented in this review for the purpose of being functional for veterinarians and animal owners.

## 2. Molecular Factors in the OA Treatment 

Many candidate genes were identified as possible targets for the OA treatment, including a wide variety of molecules such as matrix metalloproteinases (MMPs), cytokines, cathepsin K, and caspases [22]. In OA, the extracellular matrix disintegrates in synovial joints, especially in the limbs, knees, and hips, as well as indicates severe pain in suffering individuals (Figure 1).

Joint and immune cells synthesize several inflammatory mediators (cytokines), including tumor necrosis factor-alpha (TNF-α) and interleukins-1 and -7, which are important players in cartilage degradation [29]. Both IL-1β and TNF-α were shown to be increased, as well as the other cytokines (e.g., IL-8, IL-6, and leukotriene inhibitory factor), proteases, and prostaglandin E2 (PGE2) [30]. For example, IL-1β was found to be released from OA cartilage along with the inducible NOS (iNOS) [31]. PGE2 has also been reported to be spontaneously released by OA suffered cartilage [32], and leukotriene-B4 was elevated in the OA tissue synovial fluid [31]. In vitro and in vivo investigations have shown that overproduction of interleukin 1β (IL-1β) in OA is a key factor in degradation and disease progression [30]. These inflammatory mediators may trigger the production of many molecular factors such as MMPs, the enzymes which can degrade all constituents of the extracellular matrix [33]. Numerous MMPs are increased in the OA by either an elevation in their synthesis or reduced action of their suppressors [29]. MMP-1 is mainly produced by synovial cells that line the joints, and MMP-13 is a product of chondrocytes that reside in cartilage. Besides, the expression of other MMPs, including MMP-2, -3 and -9, is raised in arthritis, and these enzymes degrade non-collagen matrix elements of the joints. The collagenases MMP-1 and MMP-13 show principal roles in OA since they are rate-limiting in the collagen degradation process [34]. Cartilage damage in OA is thought to be mediated by the MMPs, which are responsible for cartilage collagen breakdown [35]. Increased levels of stromelysin (MMP-3), collagenases (MMP-1, -8, and -13), along with the gelatinases (MMP-2 and -9), were reported in the OA chondrocytes or the articular cartilage surface [29,36]. Pro-inflammatory cytokines, such as IL-1, -17, -18, and TNF-α could promote the synthesis of MMPs, while reducing the MMP enzyme inhibitors, along with the extracellular matrix synthesis [29]. Also, a previous study has reported the positive effects of using diverse in vivo gene therapy strategies with IL-1Ra in OA [37]. It was revealed that licofelone, a drug that can block both the cyclooxygenase (COX) and 5-lipoxygenase (5-LOX) pathways, efficiently reduced the development of OA structural changes while simultaneously decreasing the synthesis of leukotriene-B4 (LTB4) and IL-1β by the OA synovium [38,39,40].

## 3. Alternative Non-Surgical Treatment Approaches for OA

The quest to find the active remedies that alleviate joint degradation, amend joint flexibility, and suppress joint pain has been compelling, and present cures for treating OA comprise acetaminophen and non-steroidal anti-inflammatory drugs (NSAIDs) [41]. In humans, arthritis has been suggested to affect a considerably relatively high ratio of the US adult population, and occur at an earlier age than formerly thought [42]. NSAIDs are the present-day gold-standard pharmaceutical cure for canines suffering OA, although NSAIDs may be the reason for gastrointestinal ulcerations as an adverse effect and are contraindicated when renal insufficiency or dehydration takes place [43]. Additional therapeutic alternatives include the corticosteroids, diacerein, along with hyaluronic acid. Certain nutraceuticals, for example, chondroitin, glucosamine, pentosane polysulfate, avocado/soybean unsaponifiables, milk protein, and green-lipped mussels, are used as add-on therapies [16,44]. Following the purpose of finding the exact remedy against OA, our study group has recently suggested that, in collagen-induced arthritic rats, the arginine–silicate–inositol complex (ASI) is effective in lowering the inflammation markers β-catenin, COX-2, IL-6, MAPK, NF-κB, p38, TNF-α, and WISP-1 levels in the joint tissue of the animals [45].

Progression of the arthritic disease leads to disability that is related to joint pain and dysfunction, and it is obvious that personalized and individualized prevention and treatment strategies are needed [26]. Because of the reason that collagen is the most ubiquitous ingredient of the articular cartilage solid phase [46], UC-II supplementation has been considered as an important treatment possibility to avoid articular cartilage damages over time while supporting the therapeutic process after the OA inception [17]. Some preventive and therapeutic agents can help to reduce or prevent the progression of OA (Table 1) [47]. 

These strategies include weight control and protection of the knee structure, knee misalignment, obesity and osteotomy, physical activity, muscle strengthening for the inhibition of OA, matrix metalloprotease inhibitors, and inhibition of cytokine. Therapeutic agents include glucosamine sulfate, chondroitin sulfate, UC-II, ASI, phytochemicals (e.g., curcumin, resveratrol), steroids, and hyaluronic acid [17,47].

## 4. UC-II and its Action Mechanism

Collagens are extracellular matrix molecules used by the cells for structural integrity and a range of further functions [48]. Numerous hypotheses were suggested to clarify the precise mechanisms by which the collagen products enhance the articular cartilage health [17]. UC-II appears to exert joint-health benefits by oral tolerance, based on pre-clinical research. Oral tolerance is an immune process the body uses to distinguish between innocuous compounds (e.g., dietary proteins, intestinal bacteria) and potentially harmful foreign invaders. It takes place in the gut-associated lymphoid tissue (GALT). The GALT is mostly made up of mesenteric lymph nodes and patches of lymphoid tissue neighboring the small intestine (Peyer’s patches) [49]. Peyer’s patches take in and screen compounds from the gut lumen and, depending on the compound, switch the body’s immune response on or off. When consumed, UC-II^®^ undenatured type II collagen is believed to be taken up by the Peyer’s patches, where it activates immune cells [50]. It transforms naive T-cells into T regulatory (Treg) cells that specifically target type II collagen. Treg cells then migrate through the circulation. When they recognize type II collagen in joint cartilage, Treg cells secrete anti-inflammatory mediators (cytokines), including the transforming growth factor-beta (TGF-beta), interleukin 4 (IL-4) and interleukin 10 (IL-10) [50,51]. This action helps reduce joint inflammation and promotes cartilage repair. Undenatured type II collagen contains active epitopes that are able to interact with Peyer’s patches and induce oral tolerance. The key structural macromolecules of the cartilage tissue in the mammals are collagen and proteoglycans (aggrecan) [2,46]. Glucosamine, hyaluronic acid, and chondroitin sulfate are vital basic natural constituents of cartilage and synovial fluid. Denatured type II collagen, by contrast, lacks these essential structural components. Preclinical studies support oral tolerance as the mode of action of UC-II^®^ undenatured type II collagen and confirm that the undenatured form of type II collagen is critical for joint-health benefits: In an animal model (mouse) of RA, only undenatured type II collagen protected against joint damage, an action attributed to oral tolerance [52]. In an animal model (rat) of RA, undenatured type II collagen provided symptom relief, an action attributed to oral tolerance and modulating inflammatory pathways [51]. In a cell study, Treg cells specific for type II collagen secreted anti-inflammatory cytokines, which play a chief role in the cells’ ability to induce oral tolerance [53]. In a cell study with human chondrocytes (cells that make up cartilage), the anti-inflammatory action of IL-10 protects against damage from tumor necrosis factor-alpha (TNF-α), a pro-inflammatory mediator elevated in osteoarthritis [54]. Clinically validated lab assays confirm active epitopes in UC-II^®^ undenatured type II collagen resist digestion and retain the undenatured 3D-structure needed to interact with Peyer’s patches and induce oral tolerance [49]. This process initiates anti-inflammatory and cartilage protective pathways that prevent the immune system from injuring its joint cartilage while promoting cartilage repair and regeneration. On the other hand, immunohistochemical staining and gene expression of proteins linked to cartilage metabolism, such as collagen type II and X, matrix metallopeptidase 13 (MMP-13), sex-determining region Y-box 9 (SOX9), and connective tissue growth factor (CCN2) expressions, were suggested to be performed in the rat models of OA [17]. A possible mechanism of action for UC-II activity is briefly summarized in Figure 2.

## 5. Basic Add-on Therapies besides UC-II

During the aging period production of glucosamine, the body’s most abundant sugar and amino acid compound in mammals slows down together with the glycosaminoglycan chondroitin sulfate [13]. At present, glucosamine and chondroitin are the two most frequently used nutraceuticals that offer pharmaceutical, therapeutic, and health benefits to both human and animal arthritis patients [17]. Present medical remedies for arthritic dogs rely upon the drugs which relieve pain and regulate the inflammation to maintain daily activity [61]. Regular use of cyclooxygenase (COX) inhibitors (NSAIDs) is connected to several adverse effects, such as gastrointestinal (GI) hemorrhage and liver and kidney dysfunction [55]. In the recent past, two frequently used FDA-approved drugs (Rimadyl and Deramaxx), which are NSAIDs and elective inhibitors of COX-II, have been suggested as a reason for serious adverse effects and their safety is not evaluated in all the ages of dogs [62,63]. While paracetamol and NSAIDs are presently validated by clinical directives to treat OA [64], this emerging proof has challenged this endorsement and demonstrated the potential for adverse effects, such as cardiovascular side effects and NSAID-induced nephrotoxicity, besides GI bleeding. Glucosamine and hyaluronic acid are naturally synthesized by the body, but can also be provided via nutrition [65]. Glucosamine and chondroitin are generally recommended by veterinarians for the treatment of osteoarthritis in dogs despite the lack of compelling scientific evidence showing clinical benefit [43]. Animals-administered high intravenous concentrations of glucosamine could be especially sensitive to its diabetogenic effects via an increase of the hexosamine synthesis in the insulin-sensitive tissues, which could be a causative factor for the diabetes induction [56,57,66]. During euglycaemic–hyperinsulinaemic clamping, the glucosamine infusion in rodents, releasing plasma glucosamine concentrations of between 800 and 1200 µmol/L, resulted in glucose intolerance and insulin insensitivity [58,59].

In several studies, no effect on fasting glucose levels or glucose tolerance observations were found in other species (rabbit and dog), and the lack of a diabetogenic effect in animal feeding studies was found consistent with the low bioavailability and lack of biological outcome on the glucose metabolism [57]. Supplementation of similar main ingredients may be valuable, particularly once there is a distressed balance amongst catabolic and anabolic processes, such as in OA. On the other hand, during the progression of OA, the chondrocytes are no longer up to completely recompense the collagen type II fibers and proteoglycans loss, even at improved rates of the synthesis [22]. OA-induced animal models in dogs, rats, rabbits, and sheep have shown that hydroxycitric acid (HA) holds pleiotropic efficacy, including the anti-angiogenic, anti-fibrotic, anti-inflammatory, and anti-apoptotic effects. For instance, HA management of rats after the joint immobilization [60] or intra-articular IL-1 injection [67] were shown to protect against cartilage degeneration, seemingly because of both anti-inflammatory and anti-apoptotic effects. The main UC-II studies on humans and the animals, as well as the safety and efficacy studies, are summarized in Table 2.

## 6. OA Prevalence in Dogs

OA is the most widespread form of the arthritis type in humans and dogs, which refers to chronic joint inflammation that is caused by the cartilage deterioration. Nearly 25 percent of the 77.2 million pet dogs in the USA are diagnosed with arthritis. Dog OA is generally thought to show a similarity to human OA in terms of anatomical similarity, disease heterogeneity, and progress as well [82]. For instance, differences in articular cartilage proteoglycans occurring in slowly progressive spontaneous dog OA are strictly matched but different from those occurring in human OA. In dogs, OA is more common than RA, and pain is the leading observation. In nearly all forms of arthritis, a loss of bone or cartilage leads to a modification in the shape of joints [83]. Eventually, proteoglycan and collagen fragments are released into the synovial fluid in this stage [13]. In the adult dog, proteoglycan turnover is quicker than estimated collagen turnover, and distinct articular cartilage proteoglycan loss is permanent, which results in joint deterioration [84]. OA is typically defined as a multifactorial illness with a resilient hereditary part and can worsen by lifestyle choices to each dog specifically, which comprise diet and exercise levels [85]. In dogs, OA is mostly defined as secondary, whereby a previous major joint aberration, including the cruciate ligament rupture or patellar luxation, is supposed to stimulate the following OA growth [86]. It is not clear what percentage of dogs grow OA secondary to these or other specific predisposing situations [87]. 

It is mostly later in a dog’s life that OA turns out to be a more important problem as it has been suggested that more than 50% of diagnosed dogs are aged at 8–13 years, and thus the condition is characteristically diagnosed when mobility is significantly affected [88]. The duration that dogs are affected by OA has not been well informed in the literature due to difficulty in identifying the exact onset of the disorder and limited accessibility of long-term cohort clinical data on confirmed cases. Even though the OA may initiate at any age, it may not be clinically diagnosed until it reaches an advanced stage with clear external clinical symptoms [89]. Moreover, even though joint deterioration may already be existing when the originating cause is identified, at this point, it may not have been recorded or encoded as OA in clinical notes yet. Long term studies have found that OA can affect a considerable amount of lifespan in some affected dogs [90].

Predictions from North America have reported that the age-specific OA prevalence ranged from 20 percent in dogs older than 1 year to 80 percent in dogs older than 8 years, depending on radiographic and clinical data [91], whereas dog OA prevalence in the reported literature shows contradictory values. In the UK dog population, estimations have ranged from 6.6 percent based on primary-care data [9] to 20 percent based on referral data [89]. In the UK, among 16,437 identified candidate OA cases, 6104 of them were checked manually and 4196 of the dogs (69%) were confirmed as cases. The estimated yearly period prevalence of OA diagnosis in dogs under primary veterinary care in the UK during the year 2013 was found to be 2.5 percent. The most often affected breed prevalence was calculated as well and the most prevalent breeds were the large dog breeds, especially golden retrievers (7.7%), labrador retrievers (6.1%), rottweilers (5.4%), and german shepherds (4.9%) [12]. Almost 20% of the domestic canine population spontaneously develop OA, which is equal to nearly 15 million dogs in the USA alone. Growing evidence showed that the OA results from companion dogs reliably help to predict the drug/supplement efficiency in humans [7]. Parallel results have been seen in human studies, with many compounds of the studies undertaken in dogs having chronic pain conditions being the same as in humans [92,93]. In dogs, chondroitin sulfate, glucosamine hydrochloride, and sulfate have been considered for their anti-OA properties, which were reported to induce glycosaminoglycan formation and aggrecan production [16]. However, besides MMP-13 being a degrading collagen, it was also found to degrade the proteoglycan molecule aggrecan, therefore playing a twin role in the matrix destruction while its selective inhibition seems to have promising therapeutic approaches [34]. These supplements were reported to exert anti-catabolic and anti-inflammatory effects via the suppression of nuclear factor κB (NF-κB) binding activity [94]. 

### UC-II Usage in Dogs

Many studies have shown that UC-II improves joint mobility, flexibility, and comfort by preventing the immune system from attacking and damaging the articular cartilage [95,96,97]. In a study to assess the clinical effectiveness and safety of UC-II, obese–arthritic dogs receiving UC-II with 1 or 10 mg of UC-II/day for 90 days demonstrated reductions in the levels of overall pain, lameness, and pain during limb manipulation after the physical exercise, with 10 mg showing a greater improvement. In the same study, no adverse effects were observed in both UC-II doses, and no vital changes in serum biochemical parameters indicated that the toleration of UC-II was good. Moreover, dogs receiving UC-II for 90 days showed an increase in physical activity levels. After the withdrawal of UC-II over 30 days, all dogs experienced a general relapse, pain during exercise-related lameness, and limb manipulation [69]. In another study, the researchers tried to assess the therapeutic effectiveness and safety of glycosylated UC-II alone or in combination with glucosamine-HCl and chondroitin sulfate in 20 arthritic dogs, which were allocated into 4 groups and orally treated for 120 days. Briefly, 10 mg of UC-II was found to be superior to glucosamine and chondroitin, while the study suggested that regular treatment of arthritic dogs using UC-II alone or in combination with glucosamine and chondroitin ameliorated the signs and symptoms of arthritis considerably better than both glucosamine and chondroitin. Moreover, maximum decreases in pain were noted following the 120 days of treatment (overall pain decrease was found as 62%; pain reduction upon limb manipulation was detected as 91%, and the decrease in exercise-associated lameness was 78%) [20]. In another research, Gupta et al. [74] conducted a study to assess the therapeutic effectiveness of UC-II alone or in combination with glucosamine hydrochloride and chondroitin sulfate on client-owned moderate arthritic dogs and to determine their tolerance and safety. For this purpose, the dogs were daily treated with placebo, 10 mg active UC-II, 2000 mg glucosamine hydrochloride + 1600 mg chondroitin sulfate, and or in UC-II combined with glucosamine–hydrochloride and chondroitin–sulfate for 150 days. A significant decrease in pain was noted in the treated dogs. However, significant rises in the quantitative ground force plate (GFP) parameters (peak perpendicular force and impulse area), which is indicative of an important reduction in discomfort with arthritis, were observed only in dogs treated with UC-II. None of the dogs in the groups showed changes in physical status or liver and kidney functions. This means that active UC-II supplementation alone (10 mg/day for 150 days) was well tolerated and increased the well-being significantly in moderately arthritic dogs [74]. Another water-soluble UC-II form also exhibited similar noteworthy efficiency in relieving pain and inflammation in collagen-induced arthritis in mice, as well as moderately arthritic dogs after 150 days of supplementation with 10 mg of dosage when compared to control dogs [98,99]. In a clinical, randomized, controlled, and prospective study [79], 60 client-owned dogs were randomly allocated to the R group (*n* = 30, robenacoxib 1 mg/kg/day) and the UC-II group (UC-II 1 tablet (40 mg)/day) for a 30 days study. Based on the data obtained from the study, there was a significant reduction in the Liverpool osteoarthritis in dogs (LOAD) and mobility scores among T0 and T30 of similar size between the two groups (R = 31.5%, UC-II = 32.7%). The researchers indicated that robenacoxib and UC-II similarly improved mobility of the dogs affected by OA. 

## 7. OA Prevalence in Horses

Spontaneous joint disease is a common clinical problem in the horse. Among joint diseases, the prevalence of osteoarthritis is high, and osteoarthritis is a frequent cause of morbidity as a result of pain that is typically involved with this disease [100]. The prevalence of OA differs depending on the disease definition and reported target population. In a cross-sectional survey of horses in the UK, it was reported at 13.9% [101]. Also, the prevalence of OA found to be 50% greater than in the horses older than 15 years and up to 80%–90% in horses over 30 years [102]. In horses, OA compromises the equine industry, not only because of the treatment costs but also as a result of reduced athletic performance. Numerous epidemiological studies have suggested the prevalence of OA disease, including reports of its great incidence (more than 80%) even in Mangalarga Marchador foals aged between 12 to 36 months [103], which demonstrates that this disorder mostly affects adult and elderly horses but can also evolve in young horses and foals as well [104,105]. This situation has been stated to be because of the premature beginning of horse exercise throughout the early periods of musculoskeletal system development or because of the extreme and/or lengthy mechanical loads on undeveloped articular cartilage, by periarticular tissues incapably developed to support strong loads in many cases [106].

The prevalence and severity of metacarpophalangeal joint osteoarthritis were studied using measurable macroscopic evaluations of joints from 50 horses of different ages. They have found that one-third of horses with 2- and 3- year olds had partial or thickness lesions in the cartilage, along with the OA. Additionally, the severity of the disorder was augmented until horses become 6 years of age. In the aforementioned study, it was required to study the factors that might make 2-year-old horses susceptible to early joint affection, comprising hereditary, nutritious, and management factors as well [106]. In horses, the competing incapability is due mainly to OA, and the most common reason for euthanasia is lameness due to joint problems [107].

### UC-II Usage in Horses

In a horse study for assessing the safety and pain reduction activity of the UC-II, six groups of arthritic animals (*n* = 5–6) were tested. The researchers designed the groups as Group I: placebo, Group II: 20 mg/day, Group III: 40 mg/day, Group IV: 80 mg/day, Group V: 120 mg/day, and Group VI: 160 mg/day of UC-II, for 150 days. After the study, while placebo OA horses exhibited no changes in arthritic conditions, Groups II and III showed slight improvement. The horses that received 80, 120, and 160 mg UC-II/day revealed significant and marked improvements in the arthritic signs and were running by the end of the study [70]. In another study, horses (*n* = 5–6) received UC-II (320, 480, or 640 mg) twice daily for the first month and once daily thereafter. A significant decrease in arthritic pain was reported in horses received all dosage of UC-II. The higher daily doses of UC-II (480 and 640 mg) delivered equal welfares, representing 480 mg/day was ideal [72]. At this dose, the overall discomfort was decreased from 100% to 12% and the discomfort on limb manipulation was decreased from 100% to 22%. In the same study, glucosamine- and chondroitin-treated groups similarly revealed an important reduction in pain compared with pretreated values, whereas the efficacy was not as much of when compared with that observed with UC-II. Indeed, UC-II at 480 or 640 mg doses were more effective than glucosamine and chondroitin in arthritic horses [72].

## 8. OA Prevalence in Cats

Cats, in contrast to most dogs, can endure severe orthopedic ailments because of their small size and natural agility. Variations to OA-affected joints in cats are usually subtle. Reduced range of joint motion is generally rare in cats in comparison to dogs [108]; however, it has been reported to be more common than it is expected [109]. The huge majority of the feline OA cases are primary OA that seen in older cats with no apparent originating factor, from time to time referred to as age-related cartilage degeneration. Secondary OA in cats can be caused by several predisposing conditions such as congenital abnormality or joint irregularity and often seem after traumatic joint damage. While the prevalence of feline osteoarthritis (OA) varies, it is possibly due to dissimilar studies that have involved varying age groups of cats [110]. The primary standard for diagnosing radiographic OA is the occurrence of osteophytes; however, radiographically typical joints can also be affected by articular cartilage pathology; thus, radiographic studies are possible to undervalue the OA prevalence [111]. Nevertheless, while the prevalence of feline OA varies between publications and is likely to be biased for numerous reasons [112,113,114], more like the recent prospective, cross-sectional studies are possible to be less biased [110,115]. In cats older than 12 years of age, a 26% radiographic prevalence of appendicular OA and a 90% prevalence of total types of degenerative joint disease were found [114]. In another study, the appendicular joints of 100 cats, no specific inclusion criteria were found, and the cats were aged above 6 years, while all the cases referred to the university clinic, mostly not for the musculoskeletal system reasons. In the same study, 61 percent of the cats had OA in not less than one joint, and 48 percent had over one joint affected [115]. Also, in the cats that were aged more than 14 years, 82 percent of them had OA in one joint in any case. Thus, the prevalence of OA in those cats was found to significantly increase with age. Per that data, a randomly selected sample of 100 cats aged up to 20 years old showed that nearly all of the cats (92%) had radiographic evidence of different types of joint disease and that 91% had as a minimum of one site of appendicular joint disorder [110].

### UC-II Usage in Cats

While treatment choices for cats with OA are limited due to their sensitivity to NSAIDs compared to dogs, a novel joint supplement containing 10 mg UC-II was tested in the cats in accordance to what was assumed to decrease inflammation related to feline OA through oral tolerance, by which the immune reaction to antigens is reduced through a chronic presentation of the antigen to the GALT. The supplement was tested to measure the level of palatability and explore the tolerability levels of UC-II. For this purpose, 33 healthy cats of both genders were given one chewable UC-II containing tablet for 40 days, and at the end of the study, the observations reported. According to the data, the majority of the cats (more than 70%) in the aforementioned study consumed the chewable UC-II-containing tablets, which were well tolerated [77].

## 9. UC-II, Safety, Efficacy, and Adverse Effects

Collagen fibrils form the structural basis of the cartilage matrix and are primarily consisted of type II collagen [46]. Collagen hydrolysate is determined by the enzymatic hydrolysis of collagenous tissues, such as bone and cartilage, and from animals such as chicken and fish, although has generally been accepted as a nontoxic food component by regulatory agencies [116,117]. The key feature of UC-II is its composition of amino acids, which provide the high levels of glycine and proline, the two essential amino acids for the stability and renewal of the cartilage tissue [118]. UC-II was reported to show intact tertiary and quaternary glycoprotein integrity, which allows the epitope recognition and hypo-responsive immune stimulation, whereas the denatured type II collagen contains no tertiary or quaternary glycoprotein integrity (Figure 3) [49]. Additionally, it has been mostly derived from chicken sternum as 40 mg of UC-II material that provides 10.4 ± 1.3 mg of native type II collagen, which was encapsulated an opaque capsule with excipients [75]. A combination of radiology and histology techniques demonstrated that treatment with UC-II limits the size of the osteophytes and potentially supports the mobility and functionality of joints [17]. The structural integrity of undenatured type II collagen as an active UC-II sample was determined by a transmission electron microscope (Figure 3), while the amount of undenatured type II was characterized by ELISA [49]. 

UC-II involved the undenatured native chicken type II collagen (collagen 263.0 mg/g, hydroxyproline 32.9 mg/g), which was produced from chicken sternum cartilage in a GMP-certified facility via a patented, low-temperature manufacturing process that ensured a specific level of UC-II collagen [17]. In a recent letter, the novel faster-produced commercially available UC-II^®^ ingredient was reported to be identical with the material used in the previously published clinical research [75,76,119]. Several undenatured type II collagen, including UC-II^®^, is a patented form of collagen with undenatured type II collagen for joint health support. It has been reported that a small amount (40 mg/day) is believed to work by inducing a process known as oral tolerance that ultimately engages the immune system in the repair of its joint cartilage [74]. Earlier studies have presented that small doses of UC-II modulate joint health in both OA and RA [49]. Marone et al. [120] also reported that UC-II has an acute oral LD50 greater than 5.000 mg/kg and an acute dermal LD50 greater than 2000 mg/kg. UC-II is categorized as a mild irritant to the skin and moderately irritating to the eye based on primary skin and eye irritation tests. UC-II did not induce mutagenicity in the bacterial reverse mutation test in five Salmonella typhimurium strains, either with or without metabolic activation [120]. Similarly, UC-II did not produce a mutagenic effect in the gene mutation test in mouse lymphoma cells either with or without metabolic activation. A 90-day dose-dependent sub-chronic toxicity study in Sprague–Dawley rats showed no pathologically significant alterations in the selected organ weights individually or as a percentage of body or brain weights for oral intake of 400 mg/kg UC-II. They also reported that no important alterations were observed in hematology and clinical chemistry. Besides, it was also reported that UC-II is nontoxic for human consumption and affirmed its status as generally recognized as safe (GRAS) food ingredient [121]. Yoshinari et al. [99] also conducted a study to determine the broad-spectrum safety of new, water-soluble, undenatured type II collagen (NEXT-II) derived from chicken sternum cartilage. They reported that the acute oral LD_50_ of NEXT-II was greater than 5000 mg/kg BW in rats, whereas single-dose acute dermal LD_50_ was greater than 2000 mg/kg BW and the primary dermal irritation index (PDII) of NEXT-II was found to be 1.8 and the skin was classified as a mild irritant. In primary eye irritation research, the maximum mean total score (MMTS) of NEXT-II was found to be 7.3 and classified as minimally irritant to the eye. Long-term safety studies were performed in dogs for 150 days, and no important alterations in blood chemistry, body weight, heart rate, and respiration rate were observed. No increase in biologically significant mutagenicity and no dose-related toxicity was reported [99]. In an experimental OA model of rats, UC-II has been shown to recover the mechanical function of the injured knee and inhibit extreme degeneration of the articular cartilage [17]. Additionally, in 90 days of toxicity study of the rabbits, no pathologically important change found in the organ weights as percentages of body or brain weights, without any substantial alteration of hematology and clinical chemistry, which verified a wide-spectrum safety profile of UC-II [120].

## 10. Conclusions

UC-II administration has been reported to be more effective than the most frequently used glucosamine and chondroitin sulfate supplements in joint health studies that were done with humans and animals. UC-II can alleviate inflammatory T-cell response and activate T-regulatory cells via its oral tolerance mechanism, which eventually may reduce the cartilage damage. While NSAIDs have been shown to induce several side effects, including GI bleeding, along with the renal and hepatic dysfunction causing problems, in the long run, it is obvious that a safe and effective therapy is needed. In order to shed light to the true mechanism of action that UC-II takes place in the articular cartilage, along with an intention to bring better insights to the joint repair mechanisms, the techniques such as immunohistochemically staining and the unchecked gene expression of peptides related to cartilage metabolism should be performed in the experimental small animal models of OA. In the current literature, UC-II has been suggested as a safe and effective supplement for joint health both for humans and animals. It has been and can further be used standalone or as an adjuvant therapy with drugs in the OA management of those suffering companion animals.

## Figures and Tables

**Figure 1 animals-10-00697-f001:**
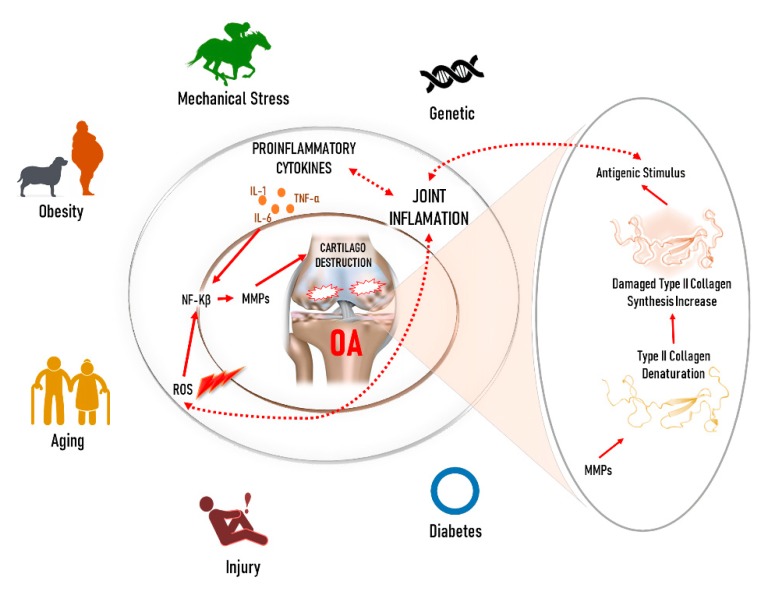
Schematic diagram of osteoarthritis; causes and inflammation mechanisms overview [23,24,25,26,27,28].

**Figure 2 animals-10-00697-f002:**
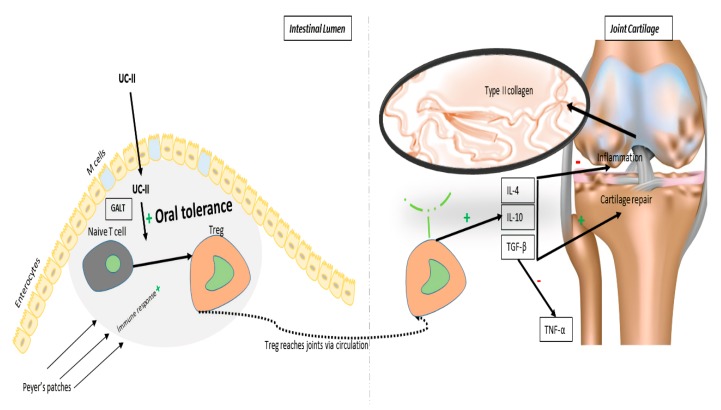
Schematic diagram of the proposed mode of action for type II collagen (UC-II) [17,55,56,57,58,59,60].

**Figure 3 animals-10-00697-f003:**
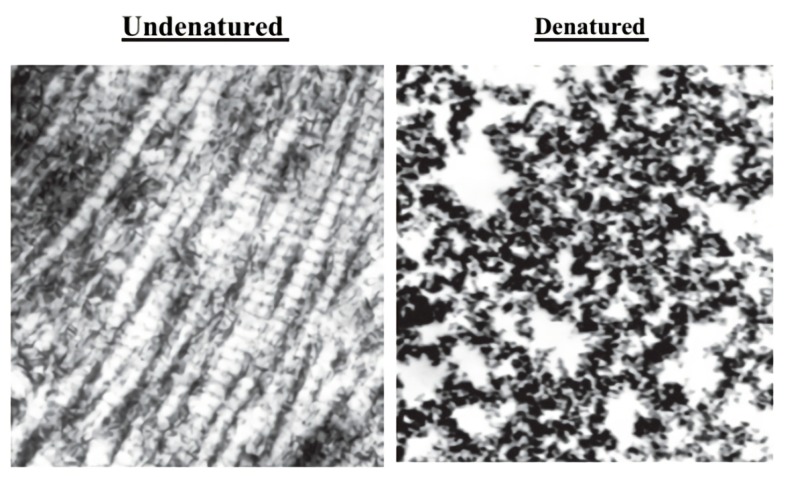
Electron micrographs of undenatured (UC-II) and denatured collagen II (×50,000) [49].

**Table 1 animals-10-00697-t001:** Main non-pharmacological and pharmaceutical preventative strategies in osteoarthritis treatments.

Non-Pharmacological and Preventative Strategies	Pharmaceutical Therapies
Weight control	NSAIDs, corticosteroids, doxycycline,
Knee misalignment and knee structure protection	MMP inhibitors
Physical rehabilitation	IL–1 receptor antagonist (IL-1Ra)
Preventing from the obesity and leptin levels management	Insulin growth factor-I (IGF-I)
Physical activity and muscle strengthening in preventing osteoarthritis	Bone anti-resorptive agents
Subchondral bone edema and bone resorption	Nutraceuticals: curcumin, EGCG, ASI
Partial meniscectomy and osteotomy	Chondroitin sulfate, glucosamine sulfate, sodium pentosan polysulfate,
Tissue engineering	Intra-articular treatments: steroids, hyaluronic acid

**Table 2 animals-10-00697-t002:** Literature overview of UC-II practice in the human and animals for osteoarthritis (OA).

No	Objective	Model	Dose and Duration	Core Findings	Conclusion	Safety	Ref.
**1**	Demonstrating the UC-II ability, whether it reduces joint pain and swelling in RA subjects.	Human	UC-II (10 mg/day) for 42 days in five female subjects (58–78 years) suffering from severe joint pain.	Reduction of pain including the stiffness was observed.	UC-II found to serve as a novel therapeutic tool in joint inflammatory conditions and symptoms of both OA and RA.	No Adverse events	[49]
**2**	Evaluating the clinical effectiveness and safety of UC-II in obese-arthritic dogs.	Dogs	Fifteen dogs in three groups received either UC-II (0 mg/day), (1 mg/day), or (10 mg/day) for 90 days, plus 30 days withdrawal.	Both UC-II receiving groups showed significant reductions in overall pain as well as pain during limb manipulation and lameness after physical exertion, also 10 mg showed better improvement. Additionally, no adverse effects and no major alterations were noted in the serum chemistry, suggesting that UC-II was well tolerated.	Daily treatment of arthritic dogs with UC-II, shown to ameliorate the signs and symptoms of arthritis. Relapse of pain was observed during the withdrawal period.	No Adverse events	[20]
**3**	Determining the therapeutic efficacy and safety of glycosylated active UC-II alone or in combination with hydroxycitric acid (HA) and chromium niacinate (CN).	Dogs	Five groups (*n* = 5) of 25 arthritic dogs received daily treatments; group I (Placebo control), group II (10 mg active UC-II), group III (1800 mg HA), group IV (1800 mg HA+100 lg CN), and group V (1800 mg HA+100 lg CN+ 10 mg active UC-II). The treatments were given for 120 days and followed up by a 30 days withdrawal period.	The dogs received the active UC-II alone (group II) or in combination (group V) for 90 days exhibited a noticeable decrease in overall, pain upon limb manipulation and exercise-related lameness. Maximum pain decrease was seen in groups II and V after 120 days of treatment. A relapse of pain was exhibited in all the dogs after 30 days of the withdrawal period.	Active UC-II was found to ameliorate the arthritic dogs alone or in combination with HA and CN. The supplements were found to be well tolerated and no adverse effects were noted.	No Adverse events	[68]
**4**	Determining the therapeutic efficacy and safety of glycosylated active UC-II alone or in combination with glucosamine-HCl and chondroitin sulfate.	Dogs	Dogs were allocated into four groups (*n* = 5), and orally treated daily for 120 days. Treatments were Group I (placebo control), Group II (10 mg UC-II), Group III (2000 mg glucosamine)+(1600 mg chondroitin sulfate), Group IV, UC-II (10 mg) + 2000 mg glucosamine + 1600 mg chondroitin sulfate, followed by a 30-day withdrawal period.	UC-II alone received dogs showed substantial reductions in overall pain within the first quarter of the study. Maximum decreases in pain were noted after 120 days of treatment. Glucosamine and chondroitin alleviated some pain, but in combination with UC-II (Group IV) significant decreases were provided in overall pain, pain upon limb manipulation and exercise-associated lameness. Following the withdrawal of supplements, all of the animals experienced a relapse of pain.	UC-II alone or in combination with glucosamine and chondroitin significantly alleviated the arthritis pain with daily treatment to the arthritic dogs, and these supplements were found to be well tolerated without any side effects.	No Adverse events	[69]
**5**	Evaluating the efficiency of pain lessening and safety of UC-II in arthritic horses.	Horses	Six groups of arthritic horses (*n* = 5–6). G. I (placebo control), G. II (UC-II 20 mg/day), G. III (UC-II 40 mg/day), G. IV (UC-II 80 mg/day), G. V (UC-II 120 mg/day), G. VI (UC-II 160 mg/day). A period of 150 days.	Groups IV, V, and VI of the horses exhibited significant improvements in the arthritic signs. Reduction in overall pain was at 79%, in pain upon limb manipulation was at 71%, and in pain, after physical exertion was at 68%. Horses receiving a higher dose of 120 and 160 mg of UC-II/day showed very little or no signs of arthritis.	UC-II at higher doses (80–160 mg/day) in the horses ameliorated the signs and symptoms of arthritis, which was also well-tolerated.	No Adverse events	[70]
**6**	Assessing the safety and efficacy of UC-II as compared to a combination of glucosamine and chondroitin (G + C) in the treatment of OA of the knee.	Human	A total of 52 subjects, half of them (*n* = 26) took a daily dose of 40 mg UC-II containing 10 mg of bioactive undenatured type II collagen via 2 capsules. Another half of the subjects (*n* = 26) took a daily dose of 1500 mg glucosamine and 1200 mg chondroitin via 4 capsules.	UC-II treatment found to be more effective when decreasing all the assessments from the baseline at 90 days. In the G + C treatment group, this effect was not observed. Specifically, although both treatments reduced the Western Ontario McMaster Osteoarthritis Index (WOMAC) score was two folds better reduced by UC-II, than the G + C treated group after 90 days.	UC-II treatment to the subjects exhibited noteworthy enhancement in daily activities, which suggested improvements for their life quality.	No Adverse events	[71]
**7**	Evaluating the arthritic pain reduction in the horses and comparison of its efficacy with the glucosamine and chondroitin	Horses	Five groups of moderate severity arthritic horses (*n* = 5–7); Group-I placebo, Group-II 320 mg UC-II, Group-III 480 mg UC-II, Group-IV 640 mg UC-II, Group-V glucosamine + chondroitin	The placebo group showed no change in arthritic conditions, whereas those receiving 320, 480, and 640 mg UC-II showed significant reductions in arthritic pain.	All supplements were tolerated well. Generally, results from this study demonstrated UC-II to be significantly more effective than the glucosamine and chondroitin supplements in arthritic horses.	No Adverse events	[72]
**8**	Assessing the safety and therapeutic effectiveness of UC-II in arthritic dogs	Dogs	Dogs were daily treated with either placebo or UC-II (10 mg active UC-II) for 120 days.	Substantial decreases (77%) were found in the overall pain of the dogs after the study period, inconsistent with pain reduction (83%) after limb manipulation and pain reduction after physical exercise (84%). Subchronic toxicity and primary dermal and eye irritation studies showed no adverse effects and UC-II did not induce mutagenic effects.	Study results demonstrated that UC-II significantly reduces arthritic pain and is safe.	No Adverse events	[73]
**9**	Determining the tolerability and safety of the therapeutic efficacy of type II collagen (UC-II) alone or in combination with glucosamine hydrochloride (GLU) and chondroitin sulphate (CHO).	Dogs	4 groups (*n* = 7–10), were treated daily with; placebo (Group-I), 10 mg active UC-II (Group-II), 2000 mg GLU + 1600 mg CHO (Group-III), and UC-II + GLU + CHO (Group-IV), for 150 days.	A significant reduction in pain was noted in Groups II, III, and IV of dogs. Significant increases in peak vertical force (N/kg body wt) and impulse area (N/kg body wt), indicative of a decrease in arthritis-associated pain, were observed in Group-II (10 mg active UC-II) dogs only. None of the dogs in any group showed changes in physical, hepatic, or renal functions.	When moderately arthritic dogs treated with UC-II (10 mg), a marked reduction in arthritic pain with maximum improvement occurred by day 150.	No Adverse events	[74]
**10**	Assessing the efficacy and tolerability of UC-II in the moderation of the joint function/pain due to strenuous exercise in healthy subjects.	Human	55 subjects who reported knee joint pain after joining in a standardized step mill performance test were randomized to take placebo (*n* = 28) or the UC-II (40 mg daily, *n* = 27) product for 120 days.	Subjects in the UC-II group showed significant improvements in average knee extension compared to placebo and to baseline. The UC-II cohort also revealed a significant change in average knee extension at day 90 versus baseline.	Daily supplementation with 40 mg of UC-II found to be well tolerated and led to improved knee joint extension. UC-II also showed the potential of increasing the period of pain-free strenuous exertion and lessen the joint pain from that.	No Adverse events	[75]
**11**	Evaluating the efficacy and safety of 150 mg of n-enriched THIAA+10 mg of UC-II in each tablet	Human	Participants took 2 tablets of nTHIAA + UC-II 2 ×/d with meals for 12 weeks.	All participants reported significant improvements in pain. The studied supplement was well tolerated, and no serious side effects occurred.	nTHIAA and UC-II were found to be safe and efficacious in participants having chronic joint pain.	No Adverse events	[50]
**12**	Evaluating the efficacy and safety of UC-II for knee OA pain and affiliated symptoms compared to glucosamine hydrochloride and chondroitin sulfate (GC).	Human	191 volunteers were randomized into three groups receiving a daily dose of UC-II (40 mg), GC (1500 mg G and 1200 mg C), or placebo for 180 days.	UC-II group demonstrated a significant reduction in overall WOMAC score compared to placebo and GC. Supplementation with UC-II also resulted in significant changes for all three WOMAC subscales. Safety outcomes did not differ among the groups.	UC-II improved knee joint symptoms in knee OA subjects and was well-tolerated.	No Adverse events	[76]
**13**	Assessing the UC-II to prevention against the excessive articular cartilage deterioration in a partial medial meniscectomy tear (PMMT) surgery performed rat model of OA.	Rats	20 male rats were used in this study. 10 rats received the vehicle and another 10 rats received an oral daily dose of UC-II at 0.66 mg/kg for 8 weeks.	PMMT surgery created a moderate OA at the medial tibia plateau. Immediate treatment with the UC-II protected the weight-bearing capacity of the injured leg, preserved the integrity of the cancellous bone at tibial metaphysis and limited the excessive osteophyte formation and deterioration of articular cartilage.	This study demonstrates that a clinically relevant daily dose of UC-II when applied immediately after an injury can improve the mechanical function of the injured knee and prevent excessive deterioration of articular cartilage.	No Adverse events	[17]
**14**	The palatability and tolerability of UC-II was studied	Cats	33 European Shorthair cats between the ages of 24 to 72 months were given one chewable tablet containing 10 mg of UC-II, daily for 40 days.	No remarkable findings on physical examination before or after the study and no appreciable changes in body weight were noted. The consumption level rose from 58% on day 0 to 73% on day 40. After an initial acquaintance period of 2–3 weeks, the level of consumption within 5 mins rose to over 70%.	10 mg of UC-II found to be very palatable in the cats studied and was well-tolerated based on physical examination.	No Adverse events	[77]
**15**	Analyzing the efficacy of UC-II alone or combined with cimicoxib, for OA treatment.	Dogs	45 dogs: 13 cimicoxib, 20 UC-II, and 12 cimicoxib + UC-II. Cimicoxib (2 mg/kg die) and UC-II tablet /day. Study lasted for 30 days.	There was a significant reduction in LOAD scores after the study. Treatment of similar magnitude among the three groups (CIMI = 31.8%, *p* < 0.001; UC-II = 32.7%, *p* = 0.013; CIM + UC-II = 31.7%, *p* = 0.009). Preliminary results of the study show similar effectiveness of the 3 treatments in reducing the degree of impairment of mobility in dogs with OA.	UC-II, while not showing a synergistic effect with cimicoxib, provided a comparable clinical efficacy to the NSAIDs itself.	No Adverse events	[78]
**16**	This study aimed to evaluate the effects of UC-II as compared to robenacoxib in OA suffering dogs.	Dogs	60 client-owned dogs were randomized in the R group (*n* = 30, robenacoxib 1 mg/kg/day for 30 days) and in the UC-II group (*n* = 30, UC-II 1 tablet/day for 30 days).	Based on the data obtained from the study, a significant reduction in LOAD and MOBILITY scores was recorded between T0 and T30 with a similar magnitude among the two groups (R = 31.5%, *p* < 0.001; UC-II = 32.7%, *p* = 0.013).	This study showed that UC-II and robenacoxib were able to similarly improve mobility of dogs affected by OA.	No Adverse events	[79]
**17**	Assessing the safety and effectiveness of un-denatured type 2 collagen in the management of OA performed in patients by 18 orthopaedicians	Human	291 patients were enrolled and followed-up at day 30 (visit 2), day 60 (visit 3), and day 90 (visit 4). Efficacy was assessed by and WOMAC and Visual Analogue scale (VAS) on each visit.	226 of 291 patients completed the 90 days study. Treatment with UC-II was related to a significant reduction in WOMAC and VAS scores.	UC-II was safe and efficacious in Indian patients having OA, which could be considered in the early management of OA.	No Adverse events	[80]
**18**	The purpose of the present study was to asses the outcome of collagen type II IN osteoarthritis of the knee joint.	Human	100 randomly selected patients that received a daily dose of UC-II (40 mg) for 120 days.	UC-II showed a significant reduction in the overall WOMAC score, LFI, and VAS scores in 120 days of observation. The UC-II led to significant changes in the three WOMAC subscales: pain *p* = 0.0005; stiffness *p* = 0.004; physical function *p* = 0.004.	UC-II improved the knee joint function in knee OA.	No Adverse events	[81]

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
