# Peer review of "Undenatured Type II Collagen (UC-II) in Joint Health and Disease: A Review on the Current Knowledge of Companion Animals"

_animals, 2020, doi:10.3390/ani10040697_

Round 1
Reviewer 1 Report
In their review, the authors report the current molecular and clinical knowledge about Osteoarthritis (OA) in animals and summarize the effects of UC-II on joint health and function in health and disease conditions in companion animals. Even if I think this review is potentially interesting for veterinary physicians and surgeons, I suggest publication after an overall revision of the manuscript.
Introduction: Undernaturated Collagen II is the main topic of this review hence I suggest to outline its properties in the introduction, so the next sections of the manuscript would be more clear and easy to read and understand. Also, I would add a sort of "aim" for the review to highlight the topic.
Sections 2 - 5: I appreciate the intense bibliographic research, but sometimes the text is confusing and distracting. For more clarity, I suggest to separate the "OA prevalence" from the therapeutic use of UC-II adding a dedicated section of the manuscript for each species considered in this review (dogs, cats and horses). I think that this kind of separation may improve the manuscript fluidity.
Figures: Figure 2 should be numbered as figure 1 and associated to the manuscript sections explaining the molecular factors leading to OA. Also, all figures should be accompanied by an explicative figure legend. Figure 3 is to blurry and the resolution should be adjusted.
The authors should slightly revise the English and correct typos. They should also revise the text formatting.
Author Response
Responses to Reviewer Comments
Reviewer 1
In their review, the authors report the current molecular and clinical knowledge about Osteoarthritis (OA) in animals and summarize the effects of UC-II on joint health and function in health and disease conditions in companion animals. Even if I think this review is potentially interesting for veterinary physicians and surgeons, I suggest publication after an overall revision of the manuscript.
Response: We thank the Reviewer for his valuable contribution.
Introduction: Undernaturated Collagen II is the main topic of this review hence I suggest to outline its properties in the introduction, so the next sections of the manuscript would be more clear and easy to read and understand. Also, I would add a sort of "aim" for the review to highlight the topic.
Response: We revised the introduction section as suggested in detail. The first paragraph was rewritten. Details of UC-II were added as suggested by the reviewer into the last paragraph of the introduction. Also, the purpose of the review has been underlined.
Sections 2 - 5: I appreciate the intense bibliographic research, but sometimes the text is confusing and distracting. For more clarity, I suggest to separate the "OA prevalence" from the therapeutic use of UC-II adding a dedicated section of the manuscript for each species considered in this review (dogs, cats and horses). I think that this kind of separation may improve the manuscript fluidity.
Response: Thanks for the suggestion. The OA prevalence and use of UC-II were organized separately as main topics according to the species at sections 6,7, and 8.
Figures: Figure 2 should be numbered as figure 1 and associated to the manuscript sections explaining the molecular factors leading to OA. Also, all figures should be accompanied by an explicative figure legend. Figure 3 is to blurry and the resolution should be adjusted.
Response: Figure numbers and their affiliations in the text were changed as suggested. Explicative figure legends added. All figure resolutions are also enhanced.
The authors should slightly revise the English and correct typos. They should also revise the text formatting.
Response: Revised. “Animal” journal template has been used to revise the text formatting.

Reviewer 2 Report
Review for Manuscript animals-750704-peer-review-v1
General Comments: The review is nicely written and provides valuable information about a new treatment/prevention for OA in veterinary patients. A few general comments:
- There is a lot of redundancy in explanation about OA after the introductory sections pathogenesis that can be trimmed/minimized to shorten the manuscript some
- When moving from human to veterinary back to human, be very careful to specify which species is being discussed, especially when it is in the same paragraph
- Check font sizes and line spacing throughout and make consistent
- More discussion should be made about surgical stabilization/arthroplasty as a treatment in the introductory passages
More specific comments are noted below by line number.
More Specific Comments:
Title - None
Abstract
- Line 15 – Change “in the animals” to “in animals”
Body
- Line 40 – Change “of the overweight” to “of overweight”
- Line 42 – Change “rheumatoid arthritis (RA)” to “immune-mediated arthritis”
- Line 52 – Specify “11th highest contributor to worldwide disability” if either dog, veterinary, or human
- Line 76 – Change “Increased ranks” to better terminology
- Line 83 – Reword “was efficiently reduces the development”
- Line 182 – Reword “healing the OA in dogs that are weak to endure the side effects”
- Line 228 – Reword “age-specific OA prevalence takes the range”
- Line 295 – Change “of late coming back to athletic performance” to “of reduced athletic performance”
Figures, Tables, and Legends – None – nicely organized.
Author Response
Responses to Reviewer Comments
Reviewer 2:
Comments and Suggestions for Authors
Review for Manuscript animals-750704-peer-review-v1
General Comments: The review is nicely written and provides valuable information about a new treatment/prevention for OA in veterinary patients.
Response: We thank the Reviewer for his valuable contribution.
A few general comments:
There is a lot of redundancy in explanation about OA after the introductory sections pathogenesis that can be trimmed/minimized to shorten the manuscript some
Response: The introduction was rewritten as suggested by the reviewer.
When moving from human to veterinary back to human, be very careful to specify which species is being discussed, especially when it is in the same paragraph
Response: Introduction was corrected as suggested.
Check font sizes and line spacing throughout and make consistent
Response: Animal Journal template has been used to revise the text formatting.
More discussion should be made about surgical stabilization/arthroplasty as a treatment in the introductory passages
Response: Additions were made in the recommended direction between the lines 54-58 of the revised manuscript.
More specific comments are noted below by line number.
More Specific Comments:
Title - None
Abstract
Line 15 – Change “in the animals” to “in animals”
Response: Corrected
Body
Line 40 – Change “of the overweight” to “of overweight”
Response: Corrected
Line 42 – Change “rheumatoid arthritis (RA)” to “immune-mediated arthritis”
Response: Corrected
Line 52 – Specify “11th highest contributor to worldwide disability” if either dog, veterinary, or human
Response: Corrected
Line 76 – Change “Increased ranks” to better terminology
Response: Corrected
Line 83 – Reword “was efficiently reduces the development”
Response: Corrected
Line 182 – Reword “healing the OA in dogs that are weak to endure the side effects”
Response: Corrected
Line 228 – Reword “age-specific OA prevalence takes the range”
Response: Corrected
Line 295 – Change “of late coming back to athletic performance” to “of reduced athletic performance”
Response: Corrected
Figures, Tables, and Legends – None – nicely organized.
Response: Thanks

Round 2
Reviewer 1 Report
The comments have been addressed nicely and the paper had a substantial improvement.
I have no other suggestion for the authors and I recommend the publication.
Author Response
Comments and Suggestions for Authors
The comments have been addressed nicely and the paper had a substantial improvement.
I have no other suggestion for the authors and I recommend the publication.
Response: We thank the Reviewer for his valuable contribution
Reviewer 2 Report
Review for Manuscript animals-750704-peer-review-v2
General Comments: The comments have been addressed nicely but just a few more specific comments below.
More specific comments are noted below by line number.
More Specific Comments:
Title – None
Abstract – None
Body
- Line 39 – Change “mobility and pain reduction” to “mobility reduction and pain”
- Line 39-42 – This passage needs to be reworded and divided into multiple sentences.
- Line 192-194 – This sentence needs to be reworded.
- Line 256 – Change “guess” to “predict”
Figures, Tables, and Legends – None
Author Response
Responses to Reviewer 2 Comments (Review for Manuscript animals-750704-peer-review-v2).
General Comments: The comments have been addressed nicely but just a few more specific comments below.
Response: We thank the Reviewer for his valuable contribution
More Specific Comments:
Line 39 – Change “mobility and pain reduction” to “mobility reduction and pain”
Response: Corrected as suggested
Line 39-42 – This passage needs to be reworded and divided into multiple sentences.
Response: Corrected as suggested
Line 192-194 – This sentence needs to be reworded.
Response: Rewritten as suggested
Line 256 – Change “guess” to “predict”
Response: Changed
Figures, Tables, and Legends – None
Response: Thanks to the reviewer
